# Effect of Various Mulch Materials on Chemical Properties of Soil, Leaves and Shoot Characteristics in *Dendrocalamus Latiflorus* Munro Forests

**DOI:** 10.3390/plants10112302

**Published:** 2021-10-26

**Authors:** Lili Fan, Ting Zhao, Muhammad Waqqas Khan Tarin, Yongzhen Han, Wenfeng Hu, Jundong Rong, Tianyou He, Yushan Zheng

**Affiliations:** 1College of Forestry, Fujian Agriculture and Forestry University, Fujian 350002, China; 2180428002@fafu.edu.cn (L.F.); zhaotingdada@163.com (T.Z.); 3200422027@fafu.edu.cn (Y.H.); 1200428007@fafu.edu.cn (W.H.); rongjd@fafu.edu.cn (J.R.); 2College of Landscape Architecture, Fujian Agriculture and Forestry University, Fujian 350002, China; waqas_tarin@yahoo.com (M.W.K.T.); heitianyou@fafu.edu.cn (T.H.)

**Keywords:** *Dendrocalamus* *latiflorus* Munro forests, mulch treatments, shooting, soil available nutrients, biochemical attributes

## Abstract

The effectiveness of mulch treatments on soil quality as well as on the yield and growth rates of bamboo are major considerations and require further attention. The present work was aimed at assessing the impacts of three different mulch materials on soil available nutrients, biochemical traits, and growth patterns of *Dendrocalamus* *latiflorus* Munro. We found that relative to the control (CK), bamboo leaves (MB) and organic fertilizers (MF) treatments significantly (*P* < 0.05) increased the number of bamboo shoots (47.5 and 22.7%) and yield (21.4 and 9.1%), respectively. We observed that under MB and MF treatments, the concentrations of soil available nutrients (nitrogen, phosphorus, and potassium) increased and played a key role in the differences in chlorophyll, leaf carbohydrate contents (soluble sugar and starch) and were essential to promote bamboo shoot development. Furthermore, we infer from principal component analysis (PCA), that both MB and MF appear to be a better choice than rice husks (MR) to improve nutrient availability, biochemical traits of the leaves, and increased bamboo shoot productivity. Consequently, we suggest using organic fertilizers and bamboo leaves as mulch materials are effective for soil conservation to attain high-quality bamboo production.

## 1. Introduction

Since the 1930s, mulch technology has been used for ecosystem modification in farmlands, forests, and urban landscapes [1]. There are several advantages to this process; mulches are known to buffer soil temperature [2], prevent evapotranspiration [3], and suppress weed germination and growth [4]. Moreover, the process can also protect soils from erosion and compaction caused by wind and water [1]. Finally, mulch can increase plant productivity by maintaining soil moisture, improving soil biological activity, and physicochemical characteristics [5]. Various mulch treatments, such as plastic film, organic materials, have diverse effects on soil physicochemical properties and plant growth [6]. Organic materials are widely used for the production of mulch materials including straw, chaff, bamboo leaves, wheat straw, organic fertilizer, etc. [7].

Bamboo is one of the key species in the world’s forest reserves, with a total area of 31.5 million hectares, accounting for about 0.8% of the world’s total forest area [8]. The mulch technology to the bamboo forest has been an innovative forest management practice, where the ground is covered with organic or inorganic materials to raise the soil temperature to ensure the early emergence of bamboo shoots and increase the yield of bamboo shoots [9]. Nevertheless, mulching on bamboo forest land may help to improve soil characteristics, enhance forest land productivity, and decrease soil erosion [10].

Various mulch materials have distinct effects on early emergence and increased yield of bamboo shoots. For instance, a greater number of *Bambusa beecheyana* var. *pubescens* (P.F.Li) W.C. Lin shoots can be produced under organic fertilizer mulches than that of bran mulches [11]. The chicken manure mulches can also significantly increase the yield of bamboo shoots [12]. In addition, the study on the coverage of *Phyllostachys praecox* prevernalis suggests that mixed mulches with chicken manure can prolong the shooting stage and significantly increase shoot yields after mulching [13,14]. The *B. oldhamii* Munro mulch with hulling has shown that bamboo shoots can be produced earlier, and their growth period can also be extended [15]. To date, studies have shown that the mulch materials such as rice husks, chicken manure, and other materials used in the *P. edulis* (Carriere) J. Houzeau forests, can promote the early emergence of bamboo shoots and increase the number of shoots and the yield [16].

Mulch can effectively improve plant production by enhancing soil quality [17]. During the decay process of organic mulch, cellulose and hemicellulose are decomposed by microorganisms, releasing nutrient elements, such as nitrogen (N), phosphorus (P), and potassium (K) into the soil [18]. As the mulching reduces the leaching of rainwater to the soil, with the obvious warming effect, the microbial activity is stimulated to improve the soil nutrient utilization and accelerate conversion efficiency [19,20]. Mulching can promote the absorption of N and P by plants and improve the quality of plants [21]. Plants absorb P to balance the increase in N into the soil, allowing them to fulfill the need for high forest productivity [22]. The increase of N under the mulch treatments can help to promote the synthesis of chlorophyll and soluble protein [23], whereas the rise of soil available P is related to the accumulation of sugar content to achieve an increase in yield [24]. Therefore, the increase of soil nutrients is the main effect of mulching [17]. Owing to the various advantages of mulch materials, the current study was designed to evaluate the response of *Dendrocalamus latiflorus* Munro using three distinct types of organic mulch materials.

*D. latiflorus* Munro is one of the most widely grown semitropical clumping bamboo species of southern China. *D. latiflorus* Munro with a high economic value can produce nutrient-rich bamboo shoots, which has great benefits to humans with increasing demand. Its bamboo products are conventionally used as the raw materials for the production of chopsticks, handicrafts, utensils, plywood, fiberboard, decorative multilayer boards, and building materials [25]. We expected that various mulch materials would have varied impacts on the chemical properties of soil, leaf biochemical traits, and growth responses of *D. latiflorus* Munro. Therefore, the present work was aimed to assess the impacts of three different mulch materials with the following objectives: (i) to examine the effects of different mulch materials on bamboo shooting characteristics and yield; (ii) to distinguish the effects of different mulch materials on leaf biochemical traits, the carbohydrate contents in the bamboo shoots, and the chemical properties of the soil; and (iii) to screen out the optimal mulch materials to maximize the shooting period. The study will provide scientific insights into cultivation management for the growth of *D. latiflorus* Munro shoot.

## 2. Results

### 2.1. Impacts of Mulch Treatments on the Number of Shoots and Bamboo Yield

Results suggested that the mulch materials increased the number of shoots and extended the period of shooting (Table 1).

Compared to the control (CK), mulch materials had a positive effect on promoting the emergence of bamboo shoots. A significant (*P* < 0.05) increase of 15, 18, and 15 days in the shooting duration was observed for bamboo leaf (MB), rice husk (MR), and organic fertilizer (MF) treatments, respectively compared to CK. In contrast, bamboo treated with MB showed an increase of 47.5% number of shoots and 21.4% shoot yield as compared to CK. Similarly, MF increased the number of shoots by 22.7% and shoot yield by 9.1% compared to CK. However, MR decreased the number of shoots (29.4 and 15.2%) and shoot yield (15.5 and 6.0%) relative to MB and MR, respectively.

### 2.2. Changes in the Soil Chemical Properties under Various Mulch Treatments

The introduction of various mulch materials (bamboo, rice, and fertilizer) also altered the dynamics of soil chemical properties; hydrolyzed nitrogen (HN), available phosphorus (AP), and available potassium (AK) (Figure 1).

Compared to CK, soil HN content greatly increased under MF and MB treatments in different phases (Figure 1A). Similarly, for AP and AK, MF treatment followed by MB showed significant (*P* < 0.05) effects over CK (Figure 1B,C). Soil HN and AK contents were greater in June and October under all mulch treatments (Figure 1A,C). Comparatively, AP was greater in June under MB and MR treatments. In October, MF treatment showed the maximum increase in soil AP relative to other mulch treatments (MB and MR) (Figure 1B).

### 2.3. Leaf Chlorophyll Contents under Various Mulch Materials

The comparisons of various mulch treatments on leaf chlorophyll contents have been summarized in Figure 2.

In June and October, compared to CK, a slight decrease in chlorophyll a/b (Chl a/b) was observed for MR and MF treatments. However, in August, compared to CK, a significant (*P* < 0.05) increase (13.0%) in Chl a/b was observed for MB, following 2.2% and 4.0% under MR and MF treatments, respectively (Figure 2A). In June and October, total chlorophyll (Tc) content increased under MB, MR, and MF treatments, respectively over CK. Moreover, in August, Tc content was not influenced under MB and MR treatments compared to CK (Figure 2B).

### 2.4. Leaf Soluble Protein Content and Carbohydrate Contents under Various Mulch Treatments

Various mulch treatments influenced leaf soluble protein and carbohydrates (Figure 3).

In June, mulch treatments did not impact the soluble protein content. However, under MB and MF treatments, the soluble protein content in leaves significantly (*P* < 0.05) increased relative to CK in August and October (Figure 3A). The soluble sugar content also increased linearly under various mulch treatments in June, August, and October. Compared to all other treatments, soluble sugar content of leaves established under MB treatment had the highest values (43.8, 62.6, and 107.8 mg g^−1^) in June, August, and October, respectively (Figure 3B). Both the leaf starch and non-structural carbohydrate (NSC) contents were significantly (*P* < 0.05) influenced under MB treatment, compared to other treatments in June and October. However, the leaf starch and NSC contents were found to decrease in all mulch treatments in August relative to the other two phases (Figure 3C,D).

### 2.5. Carbohydrate Contents in Bamboo Shoots under Various Mulch Treatments

Compared to CK, mulch treatments had a greater effect on the carbohydrate contents in the bamboo shoots, which contrasted sharply with the leaf carbohydrates (Figure 4).

MB treatment significantly (*P* < 0.05) affected soluble sugar content of shoots in June and August compared to CK. However, compared to all other mulch treatments, only MF treatment significantly (*P* < 0.05) influenced the soluble sugar content in October (Figure 4A). In all phases, the starch and NSC contents in bamboo shoots were increased under all mulch treatments relative to CK. Overall, both starch and NSC contents in bamboo shoots were greater in August under all mulch treatments (Figure 4B,C).

### 2.6. Principal Component Analysis (PCA) among Bamboo Shoot Indexes, Leaf Physiological Characteristics, and Soil Chemical Properties

The visual representation of PCA revealed that the cumulative variance contribution rate of the first two principal components reached 91.3%, which could explain all the variations of the data (Figure 5).

Among them, the first principal component can explain 59.8% of the total variation. Except for Chl a/b, other bamboo shoot indexes, leaf physiological characteristics, and soil chemical properties had positive correlations with PC1, where MB and CK differed greatly. MB showed highly positive correlations with Tc, Lsp, Bs, BNSC, AK, Bss, HN, and AP. MF and MB were positively and negatively correlated to PC2, respectively, and they had large differences in Lss, LNSC, Ls, By, and Bq, respectively, which had highly positive correlations with PC2. CK was negatively correlated with all indexes.

## 3. Discussion

According to PCA, both MB and MF treatments seem to be more advantageous than MR in terms of improving soil nutrient availability, biochemical characteristics of the leaves and shoots, and bamboo productivity. These findings are associated with severe changes owing to mulching, for instance, mulching has been shown to reduce soil surface temperatures by releasing water vapor via evapotranspiration, conserving soil moisture, retaining soil fertility, and facilitating plant development to enhance their productivity and quality [26,27]. As a result, our research provided a new perspective for studying different mulch materials to increase the yield and quality of bamboo shoots, improve soil fertility and leaf physiological status, and optimize bamboo cultivation techniques.

Mulching with bamboo leaves (MB) and organic fertilizer (MF) had substantial effects on optimizing bamboo shooting duration and increasing bamboo yield (Table 1). In comparison to the MB and MF treatments, MR had a minimal impact on improving shoot yield, which exhibited a substantial reduction. The current research findings are consistent with the conclusion of the previous research, using bamboo leaf mulch in *P. praecox Prevernalis* and *P. edulis* (Carriere) J. Houzeau [28,29,30]. The increase in the number of bamboo shoots and yield under MB and MF treatments exhibited strong correlations with the accumulation of more chlorophylls and leaf NSC contents (Figure 5). This may regulate the photosynthesis mechanism and promote the carbohydrate accumulation in *D. latiflorus* Munro to fulfill the needs of a larger number of bamboo shoots [31,32].

The plants with higher chlorophyll contents have a greater potential for carbon assimilation [33,34]. In our study, Tc content under all mulch treatments was significantly higher than CK in June and October but lower under MB and MR treatments than CK in August (Figure 2B), possibly because the decrease in temperature difference between day and night leads to enhance leaf respiration [35]. Furthermore, soluble sugar is a component of plant photosynthesis apparatus that can be used and transferred directly [36], whereas starch is a relatively long-term storage form of plant non-structural carbohydrates for soluble sugars [37]. We found that mulch treatments had significant impacts on both the soluble sugar and starch contents in leaves and promoted the leaf growth at the shoot developmental stage compared to CK, suggesting that *D. latiflorus* Munro had enhanced NSC transport ability under mulch treatments [38].

The high leaf soluble sugar content under MB treatment can ensure the increase of substances transported to the bamboo berry organs and increase the growth rate of shoots, which has been confirmed by the research of soybeans [39,40]. In addition, previous studies have shown that leaf starch (source) can decompose soluble sugars and transfer them to the organ shoots (sink) to promote shoot emergence [41]. Compared to CK, the leaf starch content under the mulch treatments declined sharply in the shooting metaphase with an increase of soluble sugar (Figure 3B,C), indicating that leaf starch decomposed soluble sugars and was imported into the organ shoots to produce a large number of bamboo shoots, which is in accordance with the change of the “source-sink” identity [42,43]. Similarly, previous studies have shown that the accumulation and conversion of abundant NSC to the rice spike is the essential mechanism to maximize the yield of rice grain [44,45]. We concluded that mulch materials significantly affected leaf growth responses of *D. latiflorus* Munro to improve the characteristics of bamboo shoots. The improved plant carbon supply and nutrient substances observed in our study could be related to bamboo leaves and organic fertilizer, which can effectively increase carbon assimilation substances in leaves and transfer them to the shoots to ensure shoot germination [39,40].

The significant impact of mulching on plant production is mostly determined by the soil nutrient concentration [17]. With the gradual release of nutrients, such as N, P, and K in the soil, and organic mulch materials, soil nutrients are absorbed and transformed by plants increasingly, and therefore the effect of mulch technology to increase productivity can be apparent [6,46]. In our research, the higher contents of HN, AP, and AK were increased under mulches (Figure 1), which is consistent with earlier research [9,13,14], demonstrating that mulches can improve the availability of soil essential nutrients stimulating the emergence of shoots and increasing yield. Soil properties may show varying effects to the different mulch treatments [6]. We found the contents of HN, AP, and AK were generally the highest under MF and MB treatments during the entire shooting period. Additionally, organic fertilizer and bamboo leaves have fast decomposition ability with a high nutrient return to soil [47,48] and can be extremely beneficial to the growth of *D. latiflorus* Munro.

In the current study, the differences in chlorophyll and bamboo carbohydrates were observed to have a positive correlation with improved soil available nutrients as a result of mulching (Figure 5), which played an important role in triggering photosynthesis activity and increasing the translocation of leaf carbohydrate to shoots [49,50]. As stated by [47], MF and MB treatments had better nutrient return to soil, and nutrients availability help to facilitate the greater carbohydrate contents and the growth of mature bamboo shoots (Figure 4). Increased AK content potentially enhances the biosynthesis of soluble protein [51]. As a result, we observed that soil available nutrients with high AK content had a substantial impact on soluble protein (Figure 5), which affected delaying plant senescence, increasing biosynthesis, and reducing the possibility of reduced bamboo yields in the lateral stages [41,50]. Previous studies have shown that the retention rate of litter decomposition about rice husk is significantly higher than other mulch materials [52]. As a consequence, the changes in soil available nutrients under rice husk mulch can help to determine the possible reasons that result in less increase of leaf Tc, soluble protein, and carbohydrate contents in bamboo shoots (Figure 2B, Figure 3A, and Figure 4C). We can infer that soil available nutrients (HN, AP, and AK) varying under various mulches could contribute to the distinct variations in leaf growth (Tc and soluble protein) and NSC accumulations in bamboo shoots, which can help to monitor the growth status of bamboo shoots during the growing phase [53].

## 4. Materials and Methods

### 4.1. Description of the Study Site

The present study was carried out in Hongxin Village in Nanjing County, Fujian Province, China. Geographically, it is situated 24°38′52″ N at latitude, 117°26′53″ E at longitude. Climatically, the study area falls in the subtropical oceanic monsoon regions of south China. The mean annual temperature is 20.4–22.3 °C with minimum and maximum temperatures of −2.9 and 40.3 °C, respectively. The mean annual frost-free period is 312 days with the mean annual precipitation of 1798 mm. The soil of the study site is terracotta with a thick humus layer (60–80 cm) with organic matter (2.07%). The basic soil physicochemical properties of the study site are as follows: pH = 4.6, total N (TN) 0.7 g·kg^−1^, total P (TP) 0.3 g·kg^−1^, total K (TK) 14.5 g·kg^−1^, HN 175.6 mg·kg^−1^, AP 8.3 mg·kg^−1^, and AK 14.6 mg·kg^−1^. The understory consists of scattered fern, herb, and shrub species, including *Dicranopteris pedata* (*Houttuyn*) *Nakaike*, *Miscanthus floridulus* (*Lab*.) *Warb*. ex *Schum et Laut*., *Rhodomyrtus tomentosa* (*Ait*.) *Hassk*. The monthly mean temperature and precipitation data during the entire study time (January to October) have been presented in Figure 6. The study site plans to replant *Musa nana Lour*. into *D. latiflorus* Munro forests over 10 years.

### 4.2. Description of Plant Material and Sampling Time

The mulching trial was conducted on 20 December 2018. Land leveling and soiling to cover the entire bamboo stump were carried out before mulching. The density of bamboo was 6–9 individuals per cluster. Three mulch materials were used: bamboo leaves, rice husks, and organic fertilizers, marked as MB, MR, and MF, and the non-covering treatment as CK. The cow dung was used as organic fertilizer. Based on the thickness of the bamboo stump being 30 cm, the organic fertilizer covered 5 kg·m^−2^, the rice husk covered 4.4 kg·m^−2^, and the bamboo leaves covered 3.1 kg m^−2^. The chemical characteristics of three mulch materials have been presented in Table 2.

We selected three plots of 25 × 25 m for this experiment. Within each plot, four treatments were arranged in a complete randomized block design representing one replicate. Furthermore, to represent the more sampling area, we also selected four bamboo clusters in the upper, middle, and lower slopes for each treatment in one plot (25 × 25 m). The average of four bamboo clusters for each treatment in one plot was considered as one replicate.

Inorganic fertilizers (N:P:K-15:15:15) were applied to all treatments in March (2 kg), June (2 kg), and August (1 kg). The fertilization method was circular furrow (the distance to bamboo strips = 20 cm). During the experiment, cultural operations such as weeding, insecticide, hooking, and irrigation were managed following normal practices.

On 18 June 2019 (shoots initial-phase), soil samples were collected to determine soil chemical properties (HN, AP, and AK) and leaf tissues were sampled to estimate the photosynthetic pigments (Chl a/b and Tc) and biochemical attributes (soluble protein, soluble sugar, and starch). The yield of fresh shoots was determined and bamboo carbohydrates (soluble sugar and starch) were also estimated. Similarly, these attributes were repeatedly measured at the second (16 August, shooting metaphase), and third phase (10 October, shooting anaphase).

### 4.3. Investigation of the Bamboo Shoots

The shooting time of each replicate was recorded from start to end of the experiment. The emergence of each replicate per treatment was calculated by the difference in the first shoot time versus CK. The number of shoots of each replicate (all treatments) were recorded with the interval of three days. The bamboo shoots were trimmed and peeled off the shell after they reached a height of approximately 1.5 m. To determine bamboo shoot yield, the edible portion of bamboo shoots was taken for weighing.

### 4.4. Analysis of Soil Chemical Properties

The soil samples were collected around bamboo stumps at four randomly selected points (20 cm in diameter, 20 cm depth) in each treatment. The soil was air-dried and sieved through a 2-mm sieve which was used for the estimation of HN, AP, and AK. HN content was measured by sodium hydroxide alkaline solution diffusion method [54], AP content was determined by molybdenum antimony colorimetric method [55], and AK content was estimated by ammonium acetate extraction-flame spectrophotometry [54]. The soil analysis was replicated four times per treatment.

### 4.5. Determination of Leaf Biochemical Attributes

During each phase, upper-middle and healthy leaves samples were collected from each replicate per treatment. Thereafter, samples were preserved with ice bags and were bought to the laboratory at Bamboo Institute of Fujian Agriculture and Forestry University. The samples were washed with distilled water and dried on filter paper.

Leaf chlorophyll contents were directly extracted from 25 mL mixed solution of acetone, absolute ethanol, and distilled water (4.5:4.5:1) as described by Gao [56] for 48–72 h in darkness until the leaves’ color changed to white completely. The absorbance of the extracted solution was measured by a UV-visible spectrophotometer (TU-1901, Beijing Puxi General Instrument Co., Ltd., Beijing, China) at the wavelengths of 645 and 663 nm. The contents of Chl a, Chl b, and Tc were calculated by using the equations of Lichtenthaler [57].

The soluble protein was determined by the chemical kit (Suzhou Keming Biotechnology Co., Ltd., Suzhou, China) of Coomassie brilliant blue method, and the absorbance of the extract was measured at 620 nm [58]. To measure soluble sugar and starch contents, a portion of fresh leaves were tagged and oven-dried at 105 °C for 15 min, later at 85 °C for drying. The dried samples were sieved to 2 mm and measured by the anthrone sulfuric acid method [59], and their absorbances were tested at the wavelength of 540 nm and 620 nm using a spectrophotometer. The NSC contents were calculated by the sum of the soluble sugar and starch contents. All leaf chemical analyses were replicated four times in each treatment.

### 4.6. Determination of Carbohydrate Content in Bamboo Shoots

During each phase, bamboo shoots with a height of 20–30 cm were collected from each replicate per treatment, preserved in ice bags, and brought back to the Bamboo Research Institute of Fujian Agriculture and Forestry University for estimation of carbohydrate contents. First, the bamboo shoots were peeled off, washed with distilled water, and dried on filter paper. Thereafter, these were chopped and placed in a kraft paper bag and oven-dried at 105 °C for 15 min, and later at 85 °C for drying. The dried samples were grinded and sieved through a 2-mm sieve. The determination method and calculation formula of starch and soluble sugar of bamboo shoots were consistent with that of leaves. The analysis was replicated four times per treatment.

### 4.7. Data Analysis

The data were statistically analyzed with SPSS-22.0 by applying one-way ANOVA per time following multiple comparison tests (LSD and Dunnett’s T3) to determine the significant differences (α 0.05). The average statistical data in three phases was adopted for PCA to analyze relationships among shoot characteristics, leaf biochemical attributes, and soil chemical properties under different treatments. Origin-lab 9.5, Prism-8.0.1, and Microsoft Excel-2016 were used for visualization and tables, respectively.

## 5. Conclusions

The findings of the present research suggest that the mulches can improve the bamboo shoot characteristics, but their effect may differ. PCA analysis revealed that mulch materials, such as MB and MF, both had a positive impact on improving the characteristics of bamboo shoots. The mulch treatments also influenced the availability of soil nutrients and biochemical characteristics of the leaves and shoots. There was a strong positive correlation between the number of shoots, the yield of shoots, and the carbohydrate contents of leaves, indicating that a large amount of accumulation of leaf photosynthetic products from leaves were transferred to bamboo shoots to promote emergence and growth. Considering the effect of various mulch materials, we conclude that both MB and MF appear to be a better choice than MR to improve bamboo productivity by increasing soil available nutrients, and biochemical traits of the leaves and shoots. Furthermore, the addition of organic fertilizer to the bamboo stands can meet the nutrient requirements of a large number of shoots during the shooting period and improve bamboo shoot quality. Therefore, we recommend the utilization of organic fertilizers and bamboo leaves as mulch materials as effective for soil conservation to attain high-quality bamboo production.

## Figures and Tables

**Figure 1 plants-10-02302-f001:**
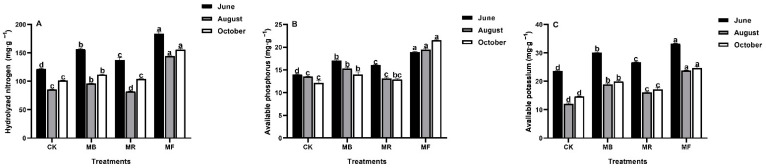
The effect of mulch treatments on soil chemical properties, where MB: bamboo leaves mulch, MR: rice husks mulch, MF: organic fertilizers mulch, and CK: control, respectively. (**A**) Hydrolyzed nitrogen (HN); (**B**) Available phosphorus (AP); (**C**) Available potassium (AK). Various letters on bars show the significant differences (*P* < 0.05) with vertical bars as standard errors (*n* = 4).

**Figure 2 plants-10-02302-f002:**
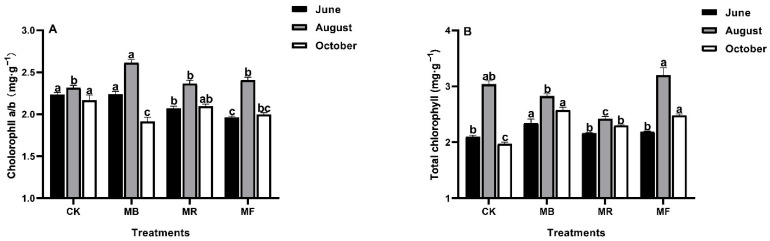
The effect of mulch treatments on leaf chlorophyll contents, where MB: bamboo leaves mulch, MR: rice husks mulch, MF: organic fertilizers mulch, and CK: control. (**A**) Chlorophyll a/b (Chl a/b); (**B**) Total chlorophyll (Tc). Various letters on each bar show significant differences (*P* < 0.05) with vertical bars as standard errors (*n* = 4).

**Figure 3 plants-10-02302-f003:**
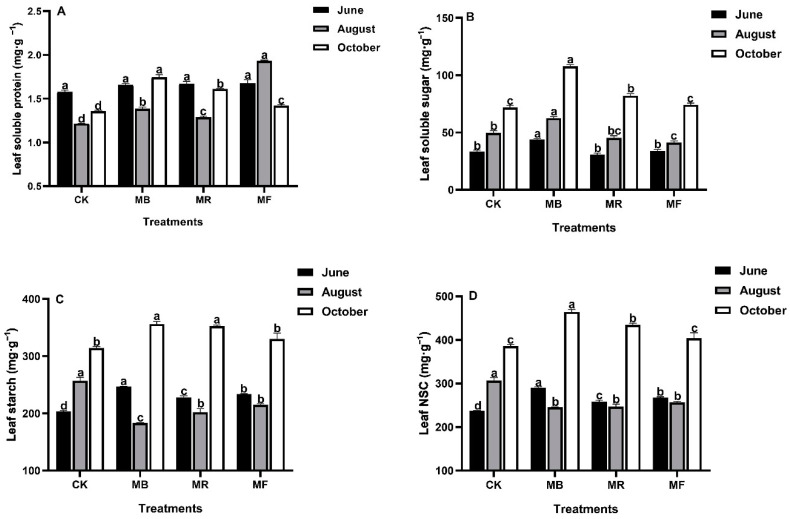
The effect of mulch treatments on leaf soluble protein content and carbohydrate contents, where MB: bamboo leaves mulch, MR: rice husks mulch, MF: organic fertilizers mulch, and CK: control. (**A**) Leaf soluble protein; (**B**) Leaf soluble sugar; (**C**) Leaf starch; (**D**) Leaf non-structural carbohydrate (NSC). Various letters on each bar show significant differences (*P* < 0.05) with vertical bars as standard errors (*n* = 4).

**Figure 4 plants-10-02302-f004:**
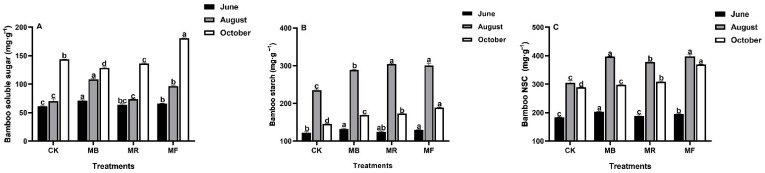
The effect of mulch treatments on carbohydrate contents in bamboo shoots, where MB: bamboo leaves mulch, MR: rice husks mulch, MF: organic fertilizers mulch, and CK: control. (**A**) Bamboo soluble sugar; (**B**) Bamboo starch; (**C**). Bamboo NSC. Various letters on each bar show significant differences (*P* < 0.05) with vertical bars as standard errors (*n* = 4).

**Figure 5 plants-10-02302-f005:**
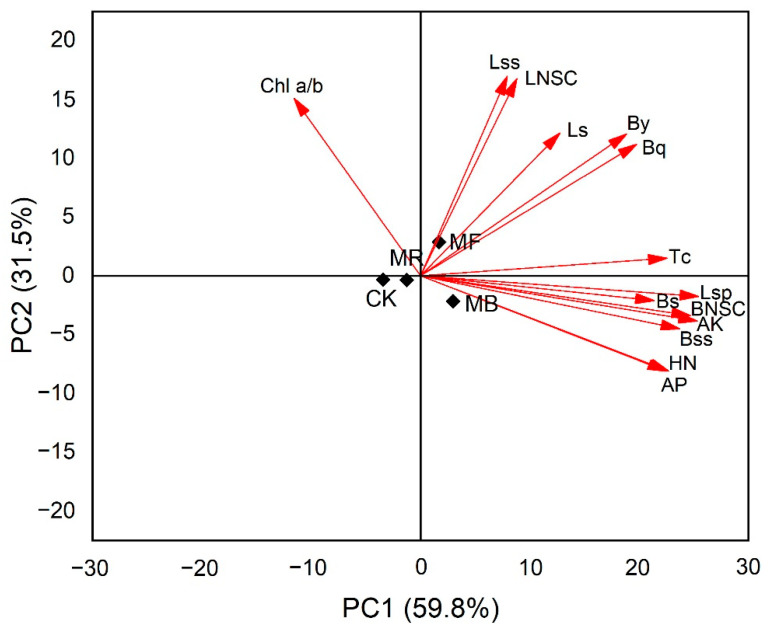
PCA of bamboo shoot indexes, leaf physiological characteristics, and soil chemical indexes, where Bq: the number of bamboo shoots; By: bamboo yield; Bss: bamboo soluble sugar; Bs: bamboo starch; BNSC: bamboo NSC; Chl a/b: chlorophyll a/b; Tc: total chlorophyll; Lsp: leaf soluble protein; Lss: leaf soluble sugar; Ls: leaf starch; LNSC: leaf NSC; HN: hydrolyzed nitrogen; AP: available phosphorus; AK: available potassium. MB: bamboo leaves mulch, MR: rice husks mulch, MF: organic fertilizers mulch, and CK: control.

**Figure 6 plants-10-02302-f006:**
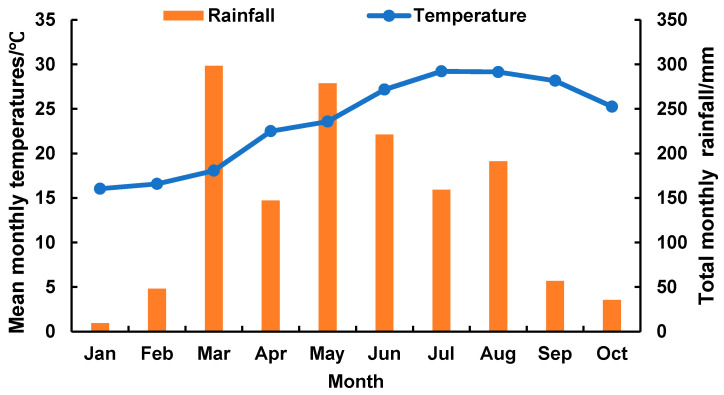
The monthly mean temperature and rainfall from 1 January to 31 October 2019.

**Table 1 plants-10-02302-t001:** The effect of mulch treatments on shooting periods and the number of shoots.

	Emergence/Days	Shooting Duration/Days	Number of Shoots/(Individual Hectare^−2^)	Bamboo Shoot Yield/(Kg Hectare^−2^)
CK	/	131.0 ± 0.9 ^b^	10,083.3 ± 659.0 ^c^	22,773.3 ± 305.5 ^a^
MB	16.0 ± 1.3 ^a^	146.0 ± 4.1 ^a^	14,875.0 ± 641.6 ^a^	27,653.3 ± 4061.1 ^a^
MR	19.0 ± 1.9 ^a^	149.0 ± 2.0 ^a^	10,500.7 ± 469.6 ^c^	23,360.0 ±1829.8 ^a^
MF	8.0 ± 1.1 ^a^	146.0 ± 3.2 ^a^	12,375.3 ± 416.7 ^b^	24,853.3 ±1649.3 ^a^

Where MB: bamboo leaves mulch, MR: rice husks mulch, MF: organic fertilizers mulch, and CK: control. Values with various letters indicate significant differences (*P* < 0.05) of mean and ± denotes the standard errors of the mean (*n* = 4).

**Table 2 plants-10-02302-t002:** The chemical characteristics of three mulch materials.

Mulch Materials	TN (g·kg^−1^)	TP (g·kg^−1^)	TK (g·kg^−1^)
Bamboo leaves	29.0 ± 0.8 ^a^	1.0 ± 0.1 ^b^	15.6 ± 0.4 ^b^
Rice husks	7.5 ± 0.2 ^b^	1.1 ± 0.1 ^b^	13.5 ± 0.3 ^c^
Organic fertilizers	28.9 ± 0.7 ^a^	4.7 ± 0.2 ^a^	25.5 ± 0.7 ^a^

Values with various letters show the significant differences (*P* < 0.05) with ± as the standard error of the mean (*n* = 3).

## Data Availability

The data presented in this study are available in the article.

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
