# Peer review of "Effect of Various Mulch Materials on Chemical Properties of Soil, Leaves and Shoot Characteristics in Dendrocalamus Latiflorus Munro Forests"

_plants, 2021, doi:10.3390/plants10112302_

Round 1

Reviewer 1 Report

See attached file

Author Response

Dear reviewer, we appreciate your comments and recommendations, which have helped us to enhance our manuscript. We have updated the text substantially given the general suggestions. For your consideration and evaluation, comments are in italic font, while responses to each concern are in red font.

Comment: (1)14 Abbreviations need to be defined (i.e. CK, MB, MF) before they are used. (2) 81 Again, the treatments need to be defined. (3) 100 Need to define the soil nutrient abbreviations.
Response: We have defined the abbreviations (i.e. CK, MB, MF, HN, AK, AP, NSC, Chl a/b) when appearing first time in the manuscript.

Comment: 33 I assume that by “suppress they growth” you mean the growth of weeds. This is not clear.
Response:
We completely agree with your suggestions and we have clearly expressed as “Suppress weed germination and growth”.

Comment: 38 remove “To date”
Response: Acknowledged and we have removed “To date”.

Comment: 52 “S.Y.Chen et C.Y.Yao suggests” something is wrong here
Response: We have revised as “Phyllostachys praecox ’Prevernalis’ suggests”.

Comment: 67 It is useful to indicate WHY you expected these changes to occur. The rationale for these hypotheses should be explained.

Response: Dear reviewer you have highlighted a valid point and we do agree with your suggestions and we have added the justification of your raised concern in the revised draft. Line # (59-72).

Comment: 86 replace “Whereas” with “However” – here and elsewhere

Response: Acknowledged.

Comment: 92 standard deviation/error of mean?? SD and SE are not the same. SE is derived from SD. So, which are you using?

Response: We have presented the mean values with SEM.

Comment: 94 Shoot days in advance/d - this unit is not at all clear; hectare is a much more common term than hectometer

Response: Dear reviewer, we have replaced “Emergence/days” with “Growth rate in days/d”, and “hectometer” with “hectare”, respectively.

Comment: 123 You do not need to point out what is in the tables or figures – refer to them parenthetically.

Response: Dear reviewer, we have revised following your suggestions.

Comment: 174 Don’t need this first sentence.

Response: Acknowledged and deleted.

Comment: 187 Comparing differences among mulches is not really possible because we don’t know the total nutrient additions in each treatment. Knowing the mass of mulch applied, multiplying by nutrient concentration, will provide that. Different mulches will have different mass per unit depth and area, so it is hard to know if differences are due to the kinds of mulches, or the nutrient content of the applied mulch.

Response: Dear reviewer we have provided the nutrient contents of all mulch material in Table2. Based on the thickness of the bamboo stump being 30 cm, the organic fertilizer covers 40 kg per cluster, the rice husk covers 35 kg per cluster, and the bamboo leaves cover 25 kg per cluster. During the shoot stage, inorganic fertilizers (N: P: K-15:15:15) were applied to all treatments in March (2 kg), June (2 kg), and August (1 kg). Besides, mulches can reduce evapotranspiration, improve soil water-holding capacity, and providing C for soil microbes. As a result, we believe that soil improvement is due to the combination of mulch and fertilizer.

Comment: 213 This sentence is incomplete, and hence confusing.

Response: Dear reviewer, we have clarified these specific lines with more detailed information, the changes are highlighted for your review and concerns. Line # (212-221).

Comment: 224 Soil nutrients “can be absorbed and transformed by plants as effective nutrients of the soil” Plants DO absorb nutrients from soil, and that is where most nutrients for most plants are derived!!

Response: We have revised the sentences, and the revised lines are highlighted for your review; Line # (229-233).

Comment: 228 Mulch likely does not directly stimulate germination and yield as this sentence suggests, but provides indirect influences by increasing soil water-holding capacity, providing C for soil microbes, and supplying nutrients.

Response: We do agree with your suggestions and we have revised the text in the revised draft, which is highlighted for your review and concerns. Line # 229-241.

Comment: 232-38 This is confusing. If mulch decomposes quickly, then how will it have a high nutrient retention capacity?

Response: Dear reviewer mulches, organic fertilizer, and bamboo leaves have fast decomposition ability and high nutrient return capacity. Besides, we have justified our findings with previous reports and changes are highlighted for your review. Line # 239-241.

Comment: 253 What is the land use history of this site?

Response: This site's land use history goes back almost ten years, when Musa nana Lour (Banana) was replaced by D. latiflorus Munro forests.

Comment: 260 “red soil” is a fairly broad description. A more in-depth taxonomic description would be helpful.
Response: The study site is located in Nanjing County, Fujian Province, China. According to Soil Science Database (http://vdb3.soil.csdb.cn/), Chinese official data indicates that the soil type of the study area is terracotta, which belongs to the subcategory of latosolic red soil. Besides we have added and highlighted the soil information in the main manuscript for your review. Line # 269.

Comment: 265 italicize scientific names

Response: Rectified.

Comment: 270 Was the site harvested of bamboo the previous year? What is the density of bamboo shoots?

Response: The bamboo shoots were harvested after 2nd year of afforestation. During the experimental period, the trial forests were planted for 10 years, therefore, bamboo shoots are harvested during entire shooting period (from June to October) every year. In the shooting anaphase, each bamboo clusters will retain 2-3 shoots to grow into mature bamboo. In the spring of each year, the density of the bamboo clusters will be adjusted with the cutting of 2-3-year-old bamboos. In general, the density of bamboo clusters is controlled at 6-9 bamboos.

Comment: 272 A description of the organic fertilizer is needed.

Response: We used cow dung used as organic fertilizer.

Comment: 274 The nutrient concentrations of mulches were provided, along with a general depth of the applied mulch. More critical, however, is knowing the actual amount of nutrients that were
applied.

Response: Based on the thickness of the bamboo stump being 30 cm, the organic fertilizer covers 40 kg per cluster, the rice husk covers 35 kg per cluster, and the bamboo leaves cover 25 kg per cluster. In addition, we have provided the nutrient contents of all mulch material in Table 2. As you stated that knowing the mass of mulch applied, multiplying by nutrient concentration, can provide nutrient contents. However, we have also determined and compared the basic soil properties at three different phases to assess the efficacy of various mulch materials.

Comment: 380 Figure S1 is not needed. This paper does not rely on seasonal precipitation or temperature patterns.

Response: Reviewer 2 recommends to show figure S1 in the section 4.1, which is relevant to introduce study site. Therefore, we want to keep this figure.

Comment: 383 This is a table – there are no vertical bars. This table should be in the actual paper, not in a supplementary table

Response: We are sorry for negligence, and we have revised the captions. Besides, the table has been placed in the revised draft as Table 2.

Comment: 384 The significant digits used here are not correct. The level of reported accuracy (3rd decimal place) is not realistic.

Response: The data presented in Table 2 is restricted up to one decimal and we also have rechecked the letters representing significance.

Reviewer 2 Report

It is an interesting research topic on forest management (mulching). The authors use appropriate data and evaluation methods. This manuscript provides an advisable and sound way to assess the effect of mulching on plant and soil chemical properties. However, the authors still should work on editing and revising the manuscript to be published.

Author Response

Dear reviewer, we appreciate your suggestions and comments, which helped to improve our article. We have revised the text significantly based on the overall comments. For your evaluation and consideration, comments are in italic font, while responses to each query are in red text.

Comment: 14-23 First time in the this section, please change it: Control (CK), bamboo leaves (MB), organic fertilizers (MF), rice husk (MR);

Response: We completely agree with your suggestions and we have changed as follows; control (CK), bamboo leaves (MB), organic fertilizers (MF), rice husk (MR). Changes are highlighted for your review in abstract.

Comment: 48-58 Replace for a number in braquets

Comment: 192, 214, 229, 233, 237, 243, 316,321,323,327,328 Please, change the references format.

Response: Dear reviewer, we are sorry for negligence, however, in the current submission, we have rectified all the citations following journal style and formatting.

Comment: 81-83 First time in this section, please write the complete names of the treatments.

Comment: 97 First time in this section, they should appear completely: Hydrolyzed nitrogen, Available phosphorus and available potassium.

Comment: 111 Chl a/b-First time in the text.

Comment: 123 NSC-First time in the text.

Response: We say sorry for the mistake and inconvenience; however, we have corrected all acronyms to full names when they appear for the first time in the manuscript, and the changes are highlighted for your review (Line # 91-44; 107-108; 121; 124; 140).

Comment: 260 It would be interesting if you add the soil classification, Luvisol? Ferrasol?.

Response: The study site is located in Nanjing County, Fujian Province, China. According to Soil Science Database (http://vdb3.soil.csdb.cn/), Chinese official data indicates that the soil type of the study area is terracotta, which belongs to the subcategory of latosolic red soil. Besides we have added and highlighted the soil classification in the main manuscript for your review (Line # 269).

Comment: 260 It would be helpful if you can provide the % of soil organic matter of this soil.

Response: Dear reviewer, sorry for our negligence, however, we have added percentage of soil organic matter of the study site in the revised draft (Line # 269).

Comment: 268 This figure it is relevant, it would be helpful if you show it in this section.

Response: The figure has been placed in the revised draft.

Comment: 272 organic fertilizers- Which ones?

Response: We applied cow dung as organic fertilizer.

Round 2

Reviewer 1 Report

Review of Effect of various mulching materials on chemical properties of soil, leaves and shoot characteristics in Dendrocalamus latiflorus Munro forests - revised

Plant Manuscript  Plants-1402805

General Comments

The authors have made substantial improvements to this paper. See below for two items that should be considered.  The writing has been improved, however additional editing, in concert with the journal editorial staff, will improve the writing even more, especially removing some wordiness. 

Specific Comments

Line        Comment

352        Express mulch addition as kg/m2, not cluster

363         Something seems amiss in TN differences.  Are the significance letters in the correct places?

Author Response

Dear reviewer, we appreciate your suggestions and comments, which helped to improve our article. We have revised the text significantly based on the overall comments. For your evaluation and consideration, comments are in italic font, while responses to each query are in red text.

Comment: 352 Express mulch addition as kg/m2, not cluster;

Response: Dear reviewer, we have expressed mulch addition as kg·m-2; Line#284-287.

Comment: 363 Something seems amiss in TN differences.  Are the significance letters in the correct places?

Response: Dear reviewer, we have reanalyzed and checked the Table 2, and the significance letters are placed correctly.